# Development and Assessment of a Six-Item Index to Gauge Motivation to Receive COVID-19 Vaccination

**DOI:** 10.3390/vaccines12010006

**Published:** 2023-12-20

**Authors:** Brian Pedersen, Gretchen Thompson, Albert Yao Kouakou, Marie Mujinga, Samuel Nicholes, Andres Martinez, Sohail Agha, Katherine Thanel, Mariame Louise Ouattara, Dorgeles Gbeke, Holly M. Burke

**Affiliations:** 1Department of Social and Behavior Change, FHI 360, Washington, DC 20009, USA; kthanel@fhi360.org; 2Department of Behavioral, Epidemiological and Clinical Sciences, FHI 360, Durham, NC 27701, USA; gthompson@fhi360.org (G.T.); snicholes@fhi360.org (S.N.); amartinez@fhi360.org (A.M.); 3Independent Research Consultant, Abidjan 00225, Côte d’Ivoire; yao_albert@ujlg.edu.ci (A.Y.K.); malououattara37@gmail.com (M.L.O.); gbekedorgeles@gmail.com (D.G.); 4Department of Social Sciences and Humanities, University of Jean Lorougnon Guédé of Daloa, Sassandra-Marahoué District, Daloa 150, Côte d’Ivoire; 5Independent Research Consultant, Kinshasa 00243, Democratic Republic of the Congo; mmujingabadibanga@gmail.com; 6Behavior Design Lab, Stanford University, Stanford, CA 94305, USA; sohailagha@gmail.com; 7Department of Reproductive, Maternal, Newborn and Child Health, FHI 360, Durham, NC 27701, USA; hburke@fhi360.org

**Keywords:** COVID-19 vaccination, vaccine hesitancy, motivation index, Fogg Behavioral Model, confirmatory factor analysis, demand creation, behavior change interventions

## Abstract

This study examined the factors influencing vaccine uptake using the Fogg Behavioral Model (FBM) and validated a multi-dimensional index for measuring a key construct in the FBM, motivation, using Confirmatory Factor Analysis (CFA) and Cronbach’s alpha. The research was conducted in Yopougon Est, Côte d’Ivoire, and Kinshasa, Democratic Republic of Congo. We aimed to develop a motivation index for COVID-19 vaccination uptake informed by the FBM. The motivation index was developed and refined using interviews and cognitive testing, and then used in baseline and endline surveys to measure the motivation to uptake COVID-19 vaccination among 2173 respondents. The index was simplified to six items validated using Confirmatory Factor Analysis (CFA) and demonstrated strong internal reliability with Cronbach’s alphas of 0.89 for the baseline and 0.77 for the endline. The study’s findings indicate that this motivation index is a valid tool for measuring motivation to receive COVID-19 vaccination, with potential applications in other vaccination campaigns. However, further testing in diverse settings is needed to enhance generalizability, including in rural areas. This research provides valuable insights for designing effective behavior change interventions to increase COVID-19 vaccination rates.

## 1. Introduction

By September 2023, over 5 billion people around the world had completed a primary series of a COVID-19 vaccine [1], and mathematical modeling has estimated that in the first year alone, nearly 20 million COVID-19-related deaths were averted [2]. Despite their demonstrable benefit, COVID-19 vaccine hesitancy increased in many regions as vaccine access improved [3,4,5]. Individual determinants associated with COVID-19 vaccine hesitancy have included the low perceived effectiveness and safety of vaccines, low perceived risk and severity of COVID-19, fear of vaccine side effects, and lack of trust in the institutions promoting vaccination [6,7,8,9,10,11,12,13,14,15,16].

Understanding and addressing the individual determinants of COVID-19 vaccination hesitancy is the essential work of behavior change interventions. To be effective, these interventions should identify and address the most significant determinants of COVID-19 vaccine hesitancy in each context. Access to timely, quality data and evidence is required for this purpose. Behavioral and social science theories and models have proven to be useful tools guiding research to identify the determinants of a behavior [17,18]. However, the accurate measurement of the constructs in these theories and models can be difficult due to their complexity and often requires validated items or indices, which can be time- and resource-intensive to develop [19].

As part of a project to increase COVID-19 vaccination rates in Yopougon Est, Abidjan, Côte d’Ivoire, and Kinshasa, Democratic Republic of Congo, we conducted research to generate evidence to design and evaluate demand creation activities [20]. Our research tools were informed by the Fogg Behavioral Model (FBM), a theoretical framework that explores the factors influencing human behavior and provides insights into designing persuasive technologies [21]. Rooted in the intersection of motivation, ability, and triggers, the model posits that behavior occurs when these three elements converge at a specific moment. Motivation reflects an individual’s desire to perform a behavior, ability denotes their capacity to do so, and triggers serve as prompts for action. The FBM has found widespread application in areas such as user experience design, health interventions, and technology development [22,23,24,25,26]. Its simplicity and emphasis on identifying optimal conditions for behavior change make it a valuable tool for designing interventions that effectively influence and facilitate the desired behaviors. The model’s adaptability and applicability across various domains underscore its significance in understanding and shaping human behavior in diverse contexts.

A validated index to measure the ability constructs of the FBM has been available since 2021 and used in COVID-19 vaccination research [20,27,28], but to our knowledge, no validated index existed to measure the motivation constructs of the FBM. The aim of our study was to develop and validate an index to measure individuals’ motivation to receive COVID-19 vaccination as characterized by the FBM.

## 2. Methods

### 2.1. Study Design and Data Collection

The motivation index described in this paper was developed as part of baseline and endline studies to inform and evaluate demand creation campaigns to increase the uptake of COVID-19 vaccination among adults living in Yopougon Est, Abidjan, Côte d’Ivoire, and Kinshasa, Democratic Republic of Congo (DRC). At the time of our research, Pfizer-BioNTech, Oxford-AstraZeneca, Sinopharm BIBP, and Janssen (J&J) were available in Yopougon Est and Janssen (J&J), Pfizer-BioNTech, and Moderna in Kinshasa. The baseline study in Yopougon Est used mixed methods, comprising key informant interviews and a quantitative survey; the baseline study in Kinshasa comprised only a quantitative survey. The endline evaluation in both locations comprised only a quantitative survey.

For key informant interviews in Yopougon Est, purposive samples were recruited from among unvaccinated and vaccinated individuals. Unvaccinated participants were identified at two community congregation points, where trained interviewers engaged with individuals who appeared to be above 18 years of age. Following age confirmation, the interviewer explained the study’s purpose and verified that the subject resided in Yopougon Est and had never received a COVID-19 vaccine. Subjects meeting these eligibility criteria were subsequently invited to participate in an interview, with their verbal consent being obtained. Similarly, vaccinated participants were identified at the post-vaccination rest area of a mass COVID-19 vaccination site, where trained interviewers engaged with individuals who appeared to be above 18 years of age. Following age confirmation, the interviewer explained the study’s purpose and verified that the subject resided in Yopougon Est. Those residing in Yopougon Est who confirmed receiving at least one dose of a COVID-19 vaccine were subsequently invited to participate in an interview, with their verbal consent being obtained.

The trained interviewers used short guides to explore the factors influencing an individual’s motivation and ability to be vaccinated and to identify interventions that might support individuals to overcome perceived barriers to COVID-19 vaccination. A total of 21 unvaccinated women and 17 unvaccinated men in the community and 10 vaccinated women and 15 vaccinated men at the mass vaccination site were interviewed in January and February 2022.

For both baseline and endline surveys, a convenience sample was recruited from among individuals entering data collection sites co-located within Internet cafés near high pedestrian trafficked areas in Yopougon Est and Kinshasa. Participants were aged 18 and above and resided in either Yopougon Est or Kinshasa. Recruitment quotas based on neighborhood, age, and gender were used to align our sample to the demographic breakdown of the commune, and we carefully timed our intercepts and selected research sites where COVID-19 vaccination demand creation campaigns were actively underway. Trained recruiters approached potential respondents entering the Internet cafés and if the individual agreed to participate, they were invited to approach a computer terminal where they would complete the survey. A structured questionnaire was used to measure vaccination status, sociodemographic characteristics, and levels of motivation and ability according to the FBM among eligible respondents. The questionnaire was the same for both baseline and endline surveys except for the addition of campaign-specific exposure questions to the endline survey.

In total, data were collected from 2173 respondents across baseline and endline surveys. Baseline survey data were collected in September 2022 in Yopougon Est, and October 2022 in Kinshasa. Endline survey data were collected in July 2023 in both Yopougon Est and Kinshasa. These studies were determined exempt by FHI 360’s Protection of Human Subjects Committee and approved by the Ivorian Ministry of Health’s National Research Ethics Committee and the University of Kinshasa, School of Public Health Ethics Committee.

### 2.2. Motivation Index Development

The motivation index was informed by the FBM, which characterizes motivation as three sets of opposing constructs—acceptance/rejection, hope/fear, and pleasure/pain [14]. Applying the FBM, we coded and analyzed responses from the key informant interviews conducted in Yopougon Est. The results of this analysis were used to develop 12 items that were aligned with each FBM motivation construct. We then assessed the items through cognitive interviews with 27 people in Yopougon Est. We assessed item comprehension, confidence in response, the level of respondent discomfort, and the accuracy in measurement by asking respondents to restate the item in their own words and comparing their response to the motivation construct of interest. The resulting 12-item index is presented in Table 1. Responses were recorded using a five-point Likert-type scale of: 1 = strongly disagree; 2 = disagree; 3 = neither agree nor disagree; 4 = agree; and 5 = strongly agree.

### 2.3. Data Analysis

We used Confirmatory Factor Analysis (CFA) in conjunction with Cronbach’s alpha to validate the motivation index based on the FBM. CFA is a form of Structural Equation Modeling (SEM) used to establish the relationship between measured variables, frequently called manifest variables, and non-measured, latent factors that serve as theoretical concepts [29,30,31]. For this study, the manifest (measured) variables are the items from the questionnaire, and the latent (non-measured) factors are the FBM motivation constructs, such as acceptance, hope, and pleasure, in addition to motivation. CFA is particularly useful in behavioral research where theoretical constructs, such as motivation, cannot be measured directly but are important to understand to inform behavior change interventions and activities [32]. In this case, we entered the FBM motivation constructs—acceptance, rejection, hope, fear, pleasure, and pain—as separate latent factors. This decision was based on the qualitative cognitive interviews used to develop our index items, as explained earlier. We conducted our analysis of baseline and endline data independently and report our results for each timepoint in the following section.

As shown in Figure 1, we constructed a second order factor model as our expected model or null hypothesis. Motivation served as the second order, latent (non-measured) factor. Acceptance, rejection, hope, fear, pleasure, and pain served as the first order, latent factors underlying motivation. Subsequently, each item from the questionnaire was influenced by one of the respective corresponding first order factors (see Figure 1). The observed data are tested against the proposed model, and fit statistics are generated to inform whether the observed data fit the model or not. Factor loadings are calculated to quantify the strength of the relationship(s) between a latent variable and the underlying items.

We also note here that Cronbach’s alpha is often used as a standard measure of internal consistency or inter-item reliability to determine whether items fit or “hang”, well together [33]. An alpha level of ≥0.70 is deemed acceptable for items and served as the threshold for this analysis. While Cronbach’s alpha is useful for determining reliability among items, it cannot be used to determine which latent factors contribute to or inform the items. CFA bridges this analytic gap by providing empirical evidence to show how items are tied together by a theoretical structure of latent factors and their respective manifest variables. All analyses were performed using R/R Studio versions 4.2.2/2023.06.2+561 [34].

## 3. Results

Table 2 shows the key characteristics of the baseline and endline survey samples that were analyzed to assess the motivation index. In total, 1161 of the respondents resided in Yopougon Est, Abidjan, and 1012 in Kinshasa, DRC. Over 40% of the respondents from both samples were under the age of 30 years. An almost perfect split between female and male participants was achieved in the baseline sample (49.87% and 50.13%, respectively), but slightly more male (55%) than female participants (45%) were surveyed in the endline sample. Both samples were highly educated with over 38% of participants reporting tertiary education. Over 55% of participants in the baseline sample and over 65% in the endline sample were employed, either part-time, full-time, or self-employed, or reported owning a business. A minority were single (15.99% of baseline participants and 17.37% of endline participants), and the vast majority (81.34% of baseline participants and 79.79% of endline participants) were Christian. In addition, 39% in the baseline sample and 59.72% in the endline sample reported having received at least one dose of a COVID-19 vaccine.

### Assessment of Motivation Index

CFA fit statistics were calculated for the baseline and endline and were found to be suboptimal for the expected model when fit against the data. Consequently, items and first order factors were trimmed to improve fit indices. Hope, rejection, pain, and their corresponding items resulted in a poor fit to the data, so we removed them from the analysis and re-fit the model without these items. This step produced a model that included all significant pathways and acceptable fit statistics, with an RMSEA of 0.074/0.040; and TLI/CFI of 0.975/0.989 and 0.990/0.996, baseline/endline, respectively (presented in Figure 2 and Table 2). Therefore, through CFA, we confirm the trimmed model as a valid arrangement when analyzing motivation using the FBM.

In our final model, Acceptance, Fear, and Pleasure and their items were retained. Factor loadings (non-standardized and standardized) and fit indices for the trimmed model are summarized in Table 3 and Table 4, respectively. As factor loadings increase, the strength of the relationship increases with an ideal magnitude of at least 1.00.

Cronbach’s alphas were generated for items retained in the six-item model to assess internal reliability and were 0.89 and 0.77 for the baseline and endline, respectively (see Table 3).

## 4. Discussion

Validated items and indices are indispensable in behavioral research for ensuring the accuracy, reliability, and construct validity of measurements. The use of validated measures facilitates comparability across studies, enhances the credibility of research, and promotes sound measurement principles. By minimizing measurement error and providing a common ground for comparison, validated indices contribute to the overall quality and trustworthiness of behavioral research findings. Furthermore, in practical applications, such as formative research, validated indices play a crucial role in making informed decisions and guiding effective interventions.

In the context of the behavioral research used to inform demand creation campaigns, inquiring directly about an individual’s motivation to adopt a behavior proves insufficient. It is imperative to discern the specific constructs that underlie motivation to determine if any require particular emphasis. Thus, we used results from key informant interviews to develop a 12-item index to measure motivation as characterized by the FBM. Each item in the index was refined by applying results from cognitive interviews, which evaluated respondent understanding, confidence, discomfort, and measurement accuracy. We then used the index in the cross-sectional baseline and endline surveys conducted in Yopougon Est, Côte d’Ivoire, and Kinshasa, DRC to inform demand creation campaigns for COVID-19 vaccination.

Using data from the cross-sectional surveys, we refined our index by using the results of a Confirmatory Factor Analysis (CFA) in conjunction with Cronbach’s alpha. Initially, CFA fit statistics for both the baseline and endline data indicated a suboptimal fit with the expected model. Consequently, we adjusted by removing hope, rejection, pain, and their associated items from the model, and negatively affected the model fit. After this refinement, the model exhibited significant pathways and satisfactory fit statistics.

The final model retained Acceptance, Fear, and Pleasure, along with their corresponding six items, as shown in Table 5. Cronbach’s alphas calculated for the retained items confirmed their internal reliability across baseline and endline surveys, suggesting that the final six-item index is reliable over time and in different populations. In a separate analysis of the data, the study team examined the association between motivation and vaccination status using logistic regression [35]. In these analyses, we find that motivation, as measured by our six-item index, is significantly associated with vaccination status in the expected direction.

Of note, our final six-item index includes items aligned with only three of the six contributory constructs that underlie motivation in the FBM. However, since the FBM characterizes motivation as a set of three opposing constructs—acceptance/rejection, hope/fear, and pleasure/pain—it might be argued that our index, which includes one side of each set of three constructs, is sufficient to measure all constructs. For example, disagreement with the acceptance item, “My family wants me to get vaccinated against COVID-19”, would indicate higher anticipated rejection. Similarly, disagreement with the pleasure item, “I would feel more at ease everyday if I were vaccinated against COVID-19”, would indicate higher anticipated pain.

Our results indicate that our index is a valid and reliable tool for measuring motivation, which can be used to inform campaigns to motivate COVID-19 vaccination uptake. Moreover, its utilization of only six items to measure motivation renders it a user-friendly tool with the potential to expedite data collection at a reduced cost. Although initially developed within the context of a demand creation campaign for COVID-19 vaccination, this index may be adaptable for understanding the motivation related to the adoption of other vaccines, particularly among adults, such as those for the influenza virus, the respiratory syncytial virus (RSV), and human papillomavirus (HPV).

### Limitations

We recognize that the sampling approaches employed in this study are non-representative, and thus limit the generalizability of our findings. Our approach was constrained by the time and resources available. Nevertheless, we took measures to address this limitation by incorporating quotas that align with critical demographic indicators, carefully timing our intercepts, and selecting research sites where COVID-19 vaccination demand creation campaigns were actively underway. These efforts served to enhance the applicability of our results within the programmatic context. Our sample size was determined as the maximum sample allowable within budgetary constraints. We do note that for inferential statistics, our sample size is well above the acceptable minimum of 300. In addition to this, we also note that in the case of our confirmatory factor analysis, we are within the accepted limits of a sample size that achieves a ratio of greater than 10 respondents for each estimated parameter or degrees of freedom [36]. It is also important to acknowledge that our study was conducted in large urban centers. We did not collect any data on populations residing in rural areas. We recommend testing our promising six-item index in rural areas and other settings seeking to increase COVID-19 vaccination rates.

## 5. Conclusions

Our findings provide a validation of a six-item index to measure motivation in the FBM and programs that use this framework. Our findings are from data collected across two urban populations spanning West and Central African countries, contributing a cross-cultural application of our findings. While further validation is needed in additional contexts—regions outside of West and Central Africa and rural contexts—our index provides researchers and program evaluators with a tested and simple approach to measuring motivation in the context of vaccination programs and emerging infectious disease.

## Figures and Tables

**Figure 1 vaccines-12-00006-f001:**
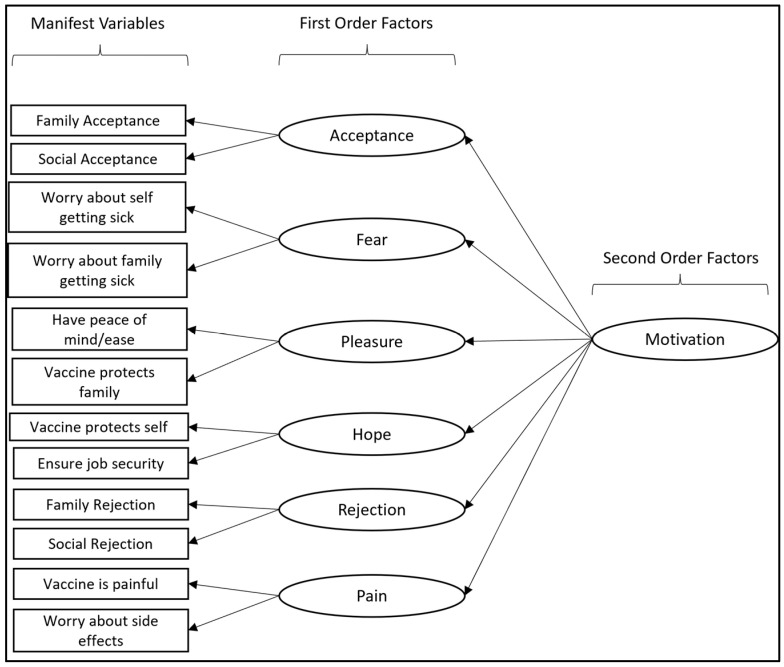
Path diagram of expected second order factor model for motivation.

**Figure 2 vaccines-12-00006-f002:**
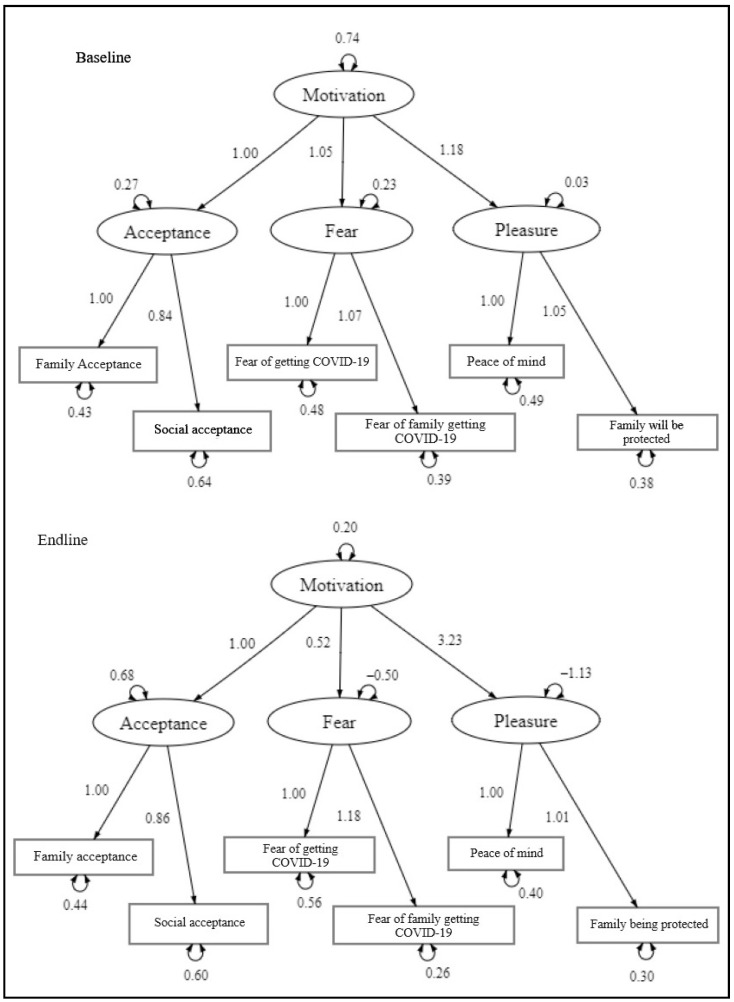
Path diagram of confirmed second order factor model for motivation.

**Table 1 vaccines-12-00006-t001:** Motivation index used in baseline and endline surveys.

Items
Acceptance
1. My family wants me to get vaccinated against COVID-19.
2. Getting vaccinated against COVID-19 would make me feel more accepted by the people around me.
Rejection
3. My family would be angry with me if I got vaccinated against COVID-19.
4. Most of the people I know would think poorly of me if I were to get a COVID-19 vaccine.
Hope
5. Getting vaccinated against COVID-19 would protect me from getting sick.
6. Getting vaccinated against COVID-19 would allow me to keep my job.
Fear
7. I worry about getting ill from COVID-19.
8. I worry about COVID-19 infecting someone in my family.
Pleasure
9. I would feel more at ease everyday if I were vaccinated against COVID-19.
10. It would make me feel good knowing that I am protecting my family by getting vaccinated against COVID-19.
Pain
11. I worry that the COVID-19 vaccine will make me sick.
12. I worry that the needlestick for the COVID-19 vaccine will be painful.

**Table 2 vaccines-12-00006-t002:** Key characteristics of the samples.

Variable	*n_baseline_*	*%*	*n_endline_*	*%*	*n_total_*
Location					
Yopougon Est, Abidjan, CI	601	54.53%	560	52.79%	1161
Kinshasa, DRC	512	45.47%	500	47.21%	1012
Age					
18–24 years	266	23.90%	213	20.09%	479
25–29 years	223	20.04%	213	20.09%	436
30–34 years	175	15.72%	140	13.21%	315
35–39 years	139	12.49%	147	13.87%	286
40–44 years	130	11.68%	137	12.92%	267
45–54 years	157	14.11%	125	11.79%	282
55+ years	21	1.89%	85	8.02%	106
Sex					
Female	555	49.87%	477	45.00%	1032
Male	558	50.13%	583	55.00%	1141
Education					
No formal education	55	4.94%	63	5.95%	118
Secondary, no diploma	94	8.45%	60	5.67%	154
Secondary	150	13.48%	162	15.30%	312
Technical training	179	16.08%	181	17.09%	360
Professional qualification	21	1.89%	36	3.40%	57
Current student, e.g., university	82	7.37%	91	8.59%	173
Tertiary	431	38.72%	412	38.90%	843
Skip/Do not know	101	9.07%	54	5.10%	155
Employment					
Unemployed	88	7.91%	76	7.18%	164
Student	159	14.29%	149	14.07%	308
Retired	19	1.71%	31	2.93%	50
Stay-at-home parent	53	4.76%	45	4.25%	98
Business owner	53	4.76%	63	5.95%	116
Independent/self-employed	169	15.18%	206	19.45%	375
Part-time	118	10.60%	165	15.58%	283
Full-time	275	24.71%	263	24.83%	538
Skip/Do not know	179	16.08%	61	5.76%	240
Marital Status					
Single	178	15.99%	184	17.37%	362
Boyfriend/girlfriend	292	26.24%	283	26.72%	575
Partnered	252	22.64%	225	21.25%	477
Married	251	22.55%	304	28.71%	555
Skip/Do not know	140	12.58%	63	5.95%	203
Religious Affiliation					
Catholic	304	27.31%	281	26.53%	585
Evangelical	231	20.75%	206	19.45%	437
Methodist	121	10.87%	122	11.52%	243
Protestant	175	15.72%	144	13.60%	319
Christian (Other)	75	6.74%	92	8.69%	167
Muslim	153	13.75%	137	12.94%	290
Traditional African Religion	33	2.96%	29	2.74%	62
Other	12	1.08%	17	1.61%	29
Skip/Do not know	9	0.81%	31	2.93%	40
Vaccination Status					
Not vaccinated	679	61.00%	427	40.28%	1106
Vaccinated	434	39.00%	633	59.72%	1067

**Table 3 vaccines-12-00006-t003:** Non-standardized and standardized loadings for a second order CFA for motivation, with Acceptance, Fear, and Pleasure as first order factors.

Second Order CFA for Motivation		Baseline		Endline	
		λ ^1^	Std	λ ^1^	Std
First Order Loadings					
Factors	Items				
Acceptance	1. My family wants me to get vaccinated against COVID-19.	1.00	0.84	1.00	0.82
	2. Getting vaccinated against COVID-19 would make me feel more accepted by the people around me.	0.84	0.72	0.86	0.72
Fear	7. I worry about getting ill from COVID-19.	1.00	0.83	1.00	0.71
	8. I worry about COVID-19 infecting someone in my family.	1.69	0.87	1.18	0.86
Pleasure	9. I would feel more at ease everyday if I were vaccinated against COVID-19.	1.00	0.83	1.00	0.84
	10. It would make me feel good knowing that I am protecting my family by getting vaccinated against COVID-19.	1.05	0.87	1.01	0.87
Second Order Loadings					
Motivation	Acceptance	1.00	0.86	1.00	0.47
	Fear	1.06	0.89	0.52	0.31
	Pleasure	1.18	0.99	3.23	1.50

^1^ Loadings were significant down to α = 0.001.

**Table 4 vaccines-12-00006-t004:** Cronbach’s alpha and fit statistics for a second order CFA for motivation.

Timepoint	Cronbach’s Alpha	CLI	TLI	RMSEA	SRMS	χ^2^	df	*p* ^1^
Baseline	0.89	0.990	0.975	0.074	0.018	42.823	6	***
Endline	0.77	0.996	0.989	0.040	0.011	16.126	6	*

^1^ Significance codes: *** < 0.001, * < 0.05.

**Table 5 vaccines-12-00006-t005:** Final motivation index.

Items
Acceptance
1. My family wants me to get vaccinated against COVID-19.
2. Getting vaccinated against COVID-19 would make me feel more accepted by the people around me.
Fear
3. I worry about getting ill from COVID-19.
4. I worry about COVID-19 infecting someone in my family.
Pleasure
5. I would feel more at ease everyday if I were vaccinated against COVID-19.
6. It would make me feel good knowing that I am protecting my family by getting vaccinated against COVID-19.

## Data Availability

De-identified aggregate data are available upon request.

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
