# Peer review of "Development and Assessment of a Six-Item Index to Gauge Motivation to Receive COVID-19 Vaccination"

_vaccines, 2023, doi:10.3390/vaccines12010006_

Round 1

Reviewer 1 Report

Comments and Suggestions for Authors

This article on the 'Development and Assessment of a Six-Item Index to Gauge Motivation to Receive COVID-19 Vaccination' represents a significant contribution to the assessment of motivation to receive COVID-19 vaccination. The authors propose a six-item index, which offers a clear and concise overview of attitudes towards vaccination at a critical stage of the pandemic.

A highlight of the analysis is the inversion of the scores of unfavourable items, a relevant methodological operation that deserves further reflection. This decision seems to have been made in order to make the favourable and unfavourable scores sum up. However, it would be desirable for the authors to provide a more thorough discussion of the implications of this inversion, both in terms of the validity of the proposed index and the interpretation of the results.

Furthermore, it might be useful to include a section that clearly and concisely summarises the rationale behind this reversal of scores, perhaps emphasising the impact on the final results and consistency with existing scientific literature. This addition would help provide the reader with a deeper understanding of the methodological choice adopted. 

Finally, it would be valuable if the authors discussed the generalisability of the results obtained through the proposed index, considering cultural and social differences that may influence motivation to vaccinate in different contexts. This aspect could further enrich the scope and applicability of the index in various populations.

In conclusion, the article represents an important step in the understanding of motivation for anti-COVID-19 vaccination, but further exploration of the methodology and its implications, together with a reflection on the generalisability of the results, could further enhance the robustness and relevance of the authors' work.

Reviewer 2 Report

Comments and Suggestions for Authors

- The aim of the study must be reported more clearly

- Sample size calculations are not given. Why did the authors recruit  1,161 of respondents resided in Yopougon and 1,012 in Kinshasa ? what is the tested hypothesis?

- The authors need to specify in much details how they recruited the participants, how they dealt with refusal to participate, whether or not an informed consent was filled.

- The authors must report why they choose the 6 areas for developing their Motivation Index. What is the rationale, what the evidence used?

- Line 135-136 These sentences need to be referenced "In this case, we entered the constructs – acceptance, rejection, hope, fear, etc. – as separate latent factors. This decision was based on the qualitative cognitive interviews used to develop our index items, as explained earlier". Avoid to use "etc."

Reviewer 3 Report

Comments and Suggestions for Authors

Nicely done.  I actually do not have any suggestions for improvement.  This is an incredibly rare event so kudos to you.

Author Response

Thank you for the comment and your time completing this review. 

Round 2

Reviewer 2 Report

Comments and Suggestions for Authors

The authors made the requested changes.

From my side the manuscript can be accepted for publication

Author Response

Thank you again for your review and comments.